# Multipole Attention for Efficient Long Context Reasoning

**Coleman Hooper**[*1]  **Sebastian Zhao**[*1]  **Luca Manolache**[1]
**Sehoon Kim**[1]  **Michael W. Mahoney**[1,2,3]  **Yakun Sophia Shao**[1]
**Kurt Keutzer**[1]  **Amir Gholami**[1,2]
[1]University of California, Berkeley  [2]ICSI  [3]LBNL
{chooper, sebbyzhao, luca.manolache, sehoonkim,
mahoneymw, ysshao, keutzer, amirgh}@berkeley.edu

## Abstract

Large Reasoning Models (LRMs) have shown promising accuracy improvements for complex problem-solving tasks. While these models have attained high accuracy by leveraging additional computation at test time, they need to generate long chain-of-thought reasoning in order to think before answering, which requires generating thousands of tokens. While sparse attention methods can help reduce the KV cache pressure induced by this long autoregressive reasoning, these methods can introduce errors which disrupt the reasoning process. Additionally, prior methods often pre-processed the input to make it easier to identify the important prompt tokens when computing attention during generation, and this pre-processing is challenging to perform online for newly generated reasoning tokens. Our work addresses these challenges by introducing MULTIPOLE ATTENTION, which accelerates autoregressive reasoning by only computing exact attention for the most important tokens, while maintaining approximate representations for the remaining tokens. Our method first performs clustering to group together semantically similar key vectors, and then uses the cluster centroids both to identify important key vectors and to approximate the remaining key vectors in order to retain high accuracy. Additionally, we design a fast cluster update process to quickly re-cluster the input and previously generated tokens, thereby allowing for accelerating attention to the previous output tokens. We evaluate our method using emerging LRMs such as Qwen-8B and Deepseek-R1-Distil-Qwen2.5-14B, demonstrating that our approach can maintain accuracy on complex reasoning tasks even with aggressive attention sparsity settings. We also provide kernel implementations to demonstrate the practical efficiency gains from our method, achieving **up to 4.5× speedup** for attention in long-context reasoning applications. Our code is available at `https://github.com/SqueezeAILab/MultipoleAttention`.

## 1  Introduction

Test-time compute has emerged as a new dimension for scaling up Large Language Model (LLM) performance [Snell et al., 2024]. By scaling the amount of computation used at test time, we can allow LLMs to think longer on harder problems in order to attain higher accuracy. Large Reasoning Models (LRMs), which are LLMs post-trained to have strong reasoning capabilities, typically reason by generating long chain-of-thought, where they produce a step-by-step chain to help guide themselves to the correct solution [Wei et al., 2022, Guo et al., 2025, Qwen, 2025]. This has been shown to

---

[*]Equal Contribution

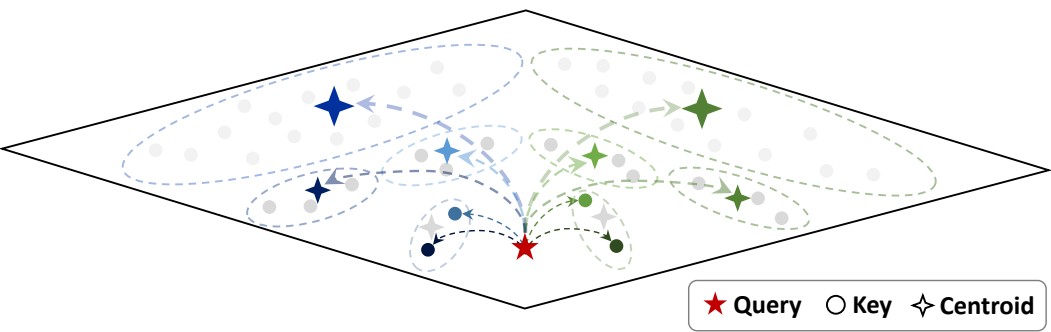

Figure 1: A visualization of the MULTIPOLE ATTENTION algorithm. For important keys (which are close to the current query), we compute exact attention. For keys which are farther away from the current query, we instead approximate the attention to these keys using the attention score with a representative cluster centroid. We use progressively coarser grained centroids as we get further away from the query. This allows us to maintain important contextual information from all tokens in the sequence, while only requiring loading a small number of keys for exact attention computation.

provide substantial accuracy gains in domains such as math and coding which require problem solving capabilities [Guo et al., 2025, Qwen, 2025].

However, the long autoregressive generation required for reasoning also brings substantial inference efficiency challenges. Notably, the reasoning process leads to large Key/Value (KV) cache memory footprint [Hooper et al., 2024b]. One common method to reduce the KV cache footprint is through sparse attention methods that only load important KV cache tokens at each generation step [Tang et al., 2024, Hooper et al., 2024a]. However, while these methods are promising for reducing the KV cache memory bottleneck, they lead to unacceptable accuracy loss. Prior work has also shown that problem solving tasks which require complex reasoning experience greater accuracy degradation when pruning the KV cache [Liu et al., 2025]. Additionally, existing sparse attention methods often pre-process the KV cache from the prompt to make it easier to retrieve important prompt tokens during generation, and this pre-processing is challenging to perform online during generation for the previously generated reasoning tokens [Hooper et al., 2024a].

To address these challenges, we build on top of [Hooper et al., 2024a] and introduce MULTIPOLE ATTENTION, which leverages sparse attention in order to only load the important KV cache tokens, while still approximating attention to the rest of the context to maintain high accuracy. Figure 1 provides an intuitive visualization of our algorithm, which first clusters the keys and computes a representative centroid for each cluster. When computing attention, our method compares the current query with the centroids to determine the importance of key tokens. For the highest scoring keys, we compute exact attention. For the remaining keys, we leverage the cluster centroid to approximate the attention score for all keys in that cluster, thereby retaining contextual information from the full KV cache. We also present a hierarchical generalization of our method which leverages progressively coarser grained centroids as we get further away from the query. To accelerate attention to the previously generated reasoning tokens, we also design a fast cluster update strategy that can be applied during generation, thereby allowing our method to accelerate attention to the previous output tokens.

Specifically, our work makes the following contributions:

- **Multipole Approximation:** Our algorithm clusters keys based on semantic similarity and represents all keys within a cluster with a representative key centroid (as in [Hooper et al., 2024a]). When computing attention, we then compare the current query with the key centroids to identify which keys are important for the current query; we only compute exact attention for these important keys, and we approximate the attention to the remaining keys using the attention scores with the cluster centroids. This allows our algorithm to maintain contextually relevant information from the full sequence, even with low KV cache budget. We also extend this to a hierarchical multipole approximation to improve the efficiency of the centroid comparison.

- **Fast Online Clustering:** In order to accelerate attention to the KV cache entries that are appended during generation, we design a fast cluster update algorithm. Our algorithm initially assigns the new tokens to clusters, and then performs a small number of refinement steps over the full keys to ensure high quality clustering. Additionally, to improve the scalability of our algorithm, we

leverage a blockwise *k*-means clustering method (which clusters the input sequence in segments), and we design a shifting window method to ensure that the final block always has sufficient tokens to perform clustering. These optimizations together facilitate efficient clustering for the generated output tokens.

- **System Implementation:** We present a prototype system implementation with custom Triton kernels for performing multipole approximate attention. Our system implementation consists of a multi-stage kernel implementation that compares the incoming user query with the key centroids, then computes exact attention for the important keys and approximate attention for the less important keys. Our methodology achieves up to **4.5×** attention speedup during decode for long context reasoning applications.

## 2 Related Work

### 2.1 Reasoning Models

Reasoning models have emerged as a new paradigm for solving complex problem-solving tasks [Guo et al., 2025, Qwen, 2025]. These models are trained to generate a chain-of-thought reasoning trajectory, which contains step-by-step reasoning to help guide the model to the correct answer [Wei et al., 2022]. The DeepSeek-R1 model [Guo et al., 2025] pioneered a new reinforcement learning-based methodology for training reasoning models for problem solving tasks. Their model release also included smaller open-source models (distilled from their base model) based on the Llama3 [Grattafiori et al., 2024] and Qwen2.5 model series [Yang et al., 2024]. There have also been several powerful reasoning models released with agentic tool-calling capabilites such as the QwQ-32B model [Qwen, 2025], the Qwen3 model series [Yang et al., 2024], and the Llama-Nemotron model series [Bercovich et al., 2025].

### 2.2 Long Context Inference

Long-context length LLMs, which can support context lengths greater than 100K tokens, have seen widespread use for applications such as document question answering, summarization, literature review, and processing multi-turn conversations. There have been several closed-source models which support long context capabilities [Achiam et al., 2023, Anthropic, 2023], including Gemini [Google, 2023] which supports greater than 1 million context length. Newer open-source model families such as Llama3 [Meta, 2024] and Qwen2.5 [Yang et al., 2024] also support greater than 100K context length. Another line of work has focused on extending the context windows of pretrained models beyond the original supported context length through scaling or adjusting the positional embeddings [Chen et al., 2023, Peng et al., 2023].

### 2.3 KV Cache Compression

KV cache compression has emerged as a key algorithmic tool for enabling long context length inference. For long context inference scenarios, the KV cache becomes the main memory bottleneck, making KV cache compression crucial for efficient inference [Hooper et al., 2024b]. Common approaches for compressing the KV cache include quantization [Hooper et al., 2024b, Liu et al., 2024b], which aims to compress the representation for each token, and sparsification [Tang et al., 2024, Hooper et al., 2024a, Zhang et al., 2024b], which aims to only load important KV cache tokens.

One previous approach for inducing sparsity to reduce KV cache size during inference is KV cache eviction, where less important tokens are evicted during the generation process to reduce memory consumption. Previous works have leveraged metrics such as attention score contribution [Zhang et al., 2024b] to select tokens to evict from the KV cache. One challenge with evicting tokens from the KV cache is that it irreversibly discards less important tokens - if these tokens become important later during generation, they cannot be recovered. KV cache merging [Wang et al., 2024, Yuan et al., 2025] was proposed as a solution to compensate for the error induced by KV cache pruning. KV cache merging involves merging evicted KV cache entries (based on key similarity [Wang et al., 2024] or historical attention scores [Yuan et al., 2025]) instead of discarding them completely. In contrast with these works, our algorithm retains the full original KV cache (thereby allowing the model to refer back to any prior KV cache state exactly if they become important during generation), while approximating attention to less important clusters using the key centroids.

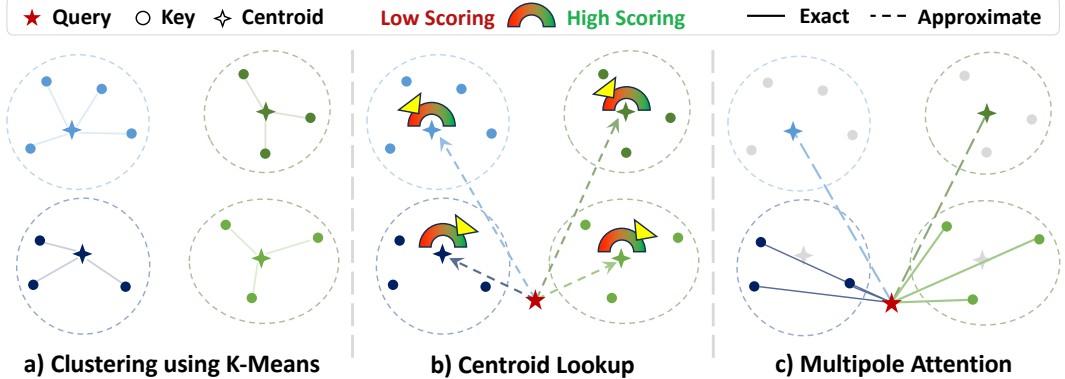

**a) Clustering using K-Means**        **b) Centroid Lookup**        **c) Multipole Attention**

Figure 2: A diagram outlining how MULTIPOLE ATTENTION identifies and retains important keys, while approximating the attention to the remaining keys. a) We first construct a Key index by performing $k$-means clustering (thereby obtaining a representative centroid for each cluster). b) When computing attention, we compare the current query with the centroids to estimate the importance of keys in each cluster ("Centroid Lookup"). c) For the most important ("High Scoring") key centroids, we then compute exact attention with the keys in the corresponding clusters. For the remaining keys, we approximate the attention to each key cluster using the attention to the representative centroid for that cluster. Overall, MULTIPOLE ATTENTION attains the efficiency of sparse attention while preserving important contextual information from the full sequence. Our method can also be generalized to a hierarchical approach, as outlined in Section 3.4.

Another common approach for leveraging sparsity for the KV cache is to retain the full KV cache and only load in the required KV cache entries for each decoding step. One prior work, QUEST [Tang et al., 2024], grouped consecutive KV cache entries and derived representative vectors for the keys within each group, and compared the query with these representative vectors to decide which KV cache entries to load. Whereas QUEST clustered keys based on positional proximity, Squeezed Attention [Hooper et al., 2024a] instead clustered keys based on semantic similarity, thereby allowing for precise identification of tokens which were likely to be high-scoring. Another related work, Tactic [Zhu et al., 2025], performed clustering based on semantic similarity and then used distribution fitting to select critical KV cache entries. Other previous works have framed the query-key comparison as a retrieval problem in order to accelerate attention using vector search methods [Zhang et al., 2024a, Liu et al., 2024a, He et al., 2025]. In contrast with existing methods, our approach aims to identify and load only a small subset of KV cache entries for exact attention computation, while approximating the attention to the remaining KV cache entries. Our algorithm clusters keys based on semantic similarity and then performs a fast centroid comparison to identify important keys, while also leveraging these representative centroids to approximate the attention to the keys which are not retrieved by the centroid lookup. A more detailed discussion of methods that sparsely load KV cache entries is provided in Appendix A.

## 3  Algorithm

### 3.1  Retrieving Important KV Cache Tokens

In order to be able to identify and retrieve important KV cache entries, we first cluster the keys based on semantic similarity using $k$-means clustering. We then derive representative centroids for each cluster by taking the mean of all vectors in each cluster, which can then be used to quickly estimate the importance of each key cluster (as in [Hooper et al., 2024a]). To identify whether the keys in cluster $i$ are important for a given query token $q$, we compare the query with the key centroid for that cluster to estimate the attention score for the corresponding keys:

$$S_i = \frac{\exp\left(qK_{c_i}^\top\right)}{\sum_j N_j \cdot \exp\left(qK_{c_j}^\top\right)}, \tag{1}$$

where $K_{c_j}$ is the key centroid for cluster $j$ and $N_j$ is the number of keys in cluster $j$. We can then use the centroid attention score to identify the clusters which are likely to be high scoring, and only load in the corresponding keys from these clusters for exact attention computation. To decide which

keys to retain, we sort the clusters based on the centroid scores, and then retain clusters until we hit a target token budget. Note that for models which support grouped query attention, we aggregate importance by averaging the estimated attention across query heads which share keys and values [Ainslie et al., 2023].

One challenge when clustering the keys is the impact of the Rotary Positional Embeddings (RoPE), which rotates key vectors by different amounts depending on their position in the sequence. This means that semantically similar keys at different positions in the sequence may not be clustered together. To address this, we employ the Windowed RoPE strategy from [He et al., 2025] when computing the centroids (which assumes a fixed relative positional difference between the query and the key vectors), thereby improving the clusterability of the key vectors. We then use a query vector rotated at a fixed position when performing the centroid lookup (as in [He et al., 2025]).

## 3.2 Importance-Aware Multipole Approximation

After using our centroid lookup to identify the important tokens, we then aim to approximate the attention to the less important tokens in order to retain necessary contextual information. We directly use the attention score to the cluster centroid from Equation 1 as the estimated attention score for the cluster. We also cache a representative value centroid $V_c$ for each cluster (which is the average of the corresponding value vectors for a given key cluster). The attention contribution for cluster $i$ (excluding the Softmax denominator) can be represented as follows:

$$N_i \exp\left(qK_{c_i}^\top\right)V_{c_i} \tag{2}$$

where $N_i$, $K_{c_i}$, and $V_{c_i}$ are the number of keys for cluster $i$, the key centroid for cluster $i$, and the value centroid for cluster $i$, respectively. For the less important key clusters, we compute the attention contribution of each cluster using Equation 2. We then merge the attention output from exact attention (for the important tokens) and from the centroids (for the less important tokens) in order to obtain the final attention output. Our algorithm is visualized in Figure 2, which shows how we leverage the key centroids both to identify important keys for the current query, as well as to approximate the attention for the less important keys.

## 3.3 Efficient Online Clustering

A key challenge when employing our approach for long generation applications is the need to approximate attention to the generated tokens. This requires updating the clusters and corresponding centroids as new tokens are appended to the KV cache; however, this is computationally expensive if we need to re-run clustering for the entire sequence. To quickly update clusters and corresponding centroids when appending newly generated tokens, we incorporate two approaches: a blockwise clustering methodology (with a sliding window to manage the transition between blocks when appending new tokens), as well as a fast initial cluster assignment for newly appended tokens followed by a small number of cluster refinement steps.

Figure 3 outlines our blockwise clustering methodology to ensure that when we append to the centroids, we do not need to recluster the full sequence. We perform clustering in blocks of $W$ tokens, which means that when we append new tokens, we only need to recluster the final block. Additionally, if the final block was limited at $W$ tokens, after exceeding this limit it would be challenging to cluster the subsequent tokens until there were a sufficient number of new tokens appended. We therefore allow the final block to extend to $\alpha + W$ tokens before removing the first $W$ tokens from this block and clustering them separately. This ensures that the final block never has fewer than $\alpha$ tokens, meaning that the final block always contains a sufficient number of keys for clustering. Note that the blockwise clustering approach also improves clustering runtimes during prefill, improving the scalability of our algorithm.

Additionally, to accelerate the update process, we sample additional initial centroids randomly from the appended tokens, and then leverage a fast initial assignment of the appended tokens to the nearest centroids and corresponding update of the centroids that these tokens map to (this is analogous to a batched version of the *sequential k*-means algorithm [MacQueen, 1967]). This process only needs to be run for the newly appended tokens, which makes its runtime overhead minimal. After running this single-shot update, to ensure high clustering accuracy we run a small number of iterations of full *k*-means over the final block to refine the centroids and centroid assignment.

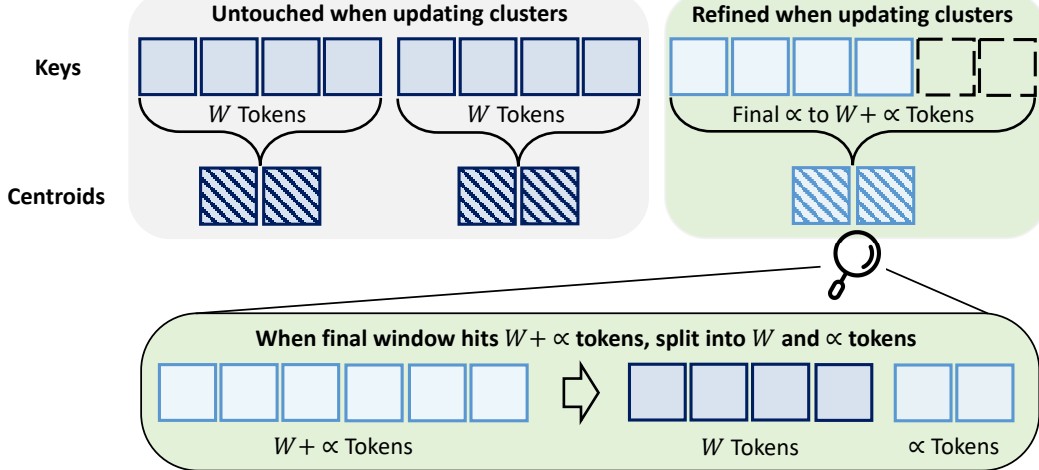

Figure 3: Visualization of our blockwise clustering method and sliding window approach for updating clusters during generation. We cluster the keys in blocks of $W$ tokens, except for the final block, which contains the last $\alpha$ to $W + \alpha$ tokens. When we update the clusters, we only need to update the final window. Once the final window reaches $W + \alpha$ tokens, we split it into the first $W$ tokens (which becomes its own block) and the final $\alpha$ tokens (which becomes the new final block that we append new keys to).

### 3.4 Hierarchical Multipole Attention

Although our method allows for pruning a large portion of KV cache entries while maintaining accuracy, the granularity of clustering needs to be sufficient for accurate retrieval and approximation. However, this can lead to the centroid lookup becoming a bottleneck if the number of centroids that we need to compare with grows with the context length. In particular, as we push to more aggressive sparsity regimes, the overhead of the centroid lookup becomes more prominent.

To alleviate the overhead of the centroid lookup, while still providing an accurate approximation for the full KV cache, we extend the hierarchical centroid lookup method from [Hooper et al., 2024a] to a hierarchical multipole approximation for the attention operation. We perform hierarchical $k$-means clustering in order to derive progressively coarser grained cluster centroids. This hierarchical clustering is compatible with the fast online clustering update described in Section 3.3, as the blockwise clustering method can be applied at each level of the hierarchy. At each level of the centroid lookup, we identify a small number of promising key centroids (for which we need to compare with the next level of refined centroids), and then approximate the less important keys using the attention to the key centroids.

For the case where we have two levels of hierarchy, we first compare the query with the coarse-grained (first-level) centroids to identify which regions of the keys are potentially important. This initial lookup allows us to only perform fine-grained (second-level) centroid comparisons with centroids that are likely to be high scoring. Additionally, for the low-importance first-level centroids, we can approximate the attention to all of the tokens in the corresponding clusters using the attention to the first-level centroids. We then perform the second level centroid lookup only for the promising fine-grained centroids. Here, we identify the most important tokens (which need to be loaded for exact attention computation), and we approximate the attention to the remaining tokens which were only important enough to reach the second-level centroid lookup. This progressive refinement process allows us to employ more accurate approximation for keys depending on their importance. Overall, our hierarchical multipole algorithm reduces the overhead of the centroid comparison, while still providing accurate approximation for the less important keys at each step in the lookup.

## 4    System Implementation

We implement custom Triton kernels for computing the centroid lookup, sparse Flash Decoding, and centroid replacement operations. These kernels follow the parallelization strategies from FlashAttention-2 [Dao, 2023], including splitting computation across attention heads and along

the query sequence length dimension, and FlashDecoding [Dao et al., 2023], where computation is split across the KV dimension and the intermediate results are reduced. We also build upon existing work on sparse FlashAttention kernels [Pagliardini et al., 2023, Hooper et al., 2024a].

The first stage of our kernel implementation (the centroid lookup) is designed to compare the query with the key centroids in order to identify the important tokens (as in [Hooper et al., 2024a]). After obtaining the attention scores to the centroids, we gather the centroid scores per-token and apply top-K to identify which top-scoring clusters of keys can be retained under a fixed token budget. After identifying the important and less important clusters, the next phase of our kernel implementation (sparse FlashDecoding) calculates the attention output from keys belonging to selected (important) clusters. This kernel accepts as input a tensor containing the indices of keys, and only loads the keys corresponding to these indices for exact attention computation.

Finally, in the third stage of our kernel implementation (centroid replacement), we approximate the attention to the less important keys using the attention to the corresponding centroids. Based upon Equation 2, we need $\exp\left(qK_{c_i}^\top\right)$, i.e. the result of comparing the query with the key centroids. Given that these values are computed during centroid lookup, we modify the centroid lookup kernels to save these values and then we later load them in the replacement kernel (to avoid re-loading the key centroids during the replacement stage). We also return a boolean mask from the centroid lookup that identifies which centroids are less important in order to only perform replacement for these centroids. The centroid replacement kernel loads the query-key centroid dot product results from the centroid lookup and then computes attention with the value centroids. The attention output is then merged with the output from the sparse FlashDecoding kernel.

For our fast clustering update, we implement this using PyTorch-level primitives (wrapped in torch.compile to reduce kernel launch overheads). We use a fixed buffer of $L$ to $2L$ local tokens which are not clustered and are loaded exactly at each decoding step, and we append the oldest $L$ tokens from the buffer to the clusters every $L$ decoding steps. After performing clustering, we cache the key and value centroids in order to avoid recomputing these values for each generation step.

## 5 Results

### 5.1 Experimental Setup

We evaluate MULTIPOLE ATTENTION on two long context reasoning datasets: LongBenchV2 [Bai et al., 2024] and GSM-Infinite [Zhou et al., 2025]. LongBenchV2 contains complex real-world long-context questions across tasks such as document QA, long in-context learning, dialogue understanding, and code repository understanding. GSM-Infinite contains synthetic mathematical reasoning tasks that require multiple arithmetic operations. We leverage two open-source reasoning models for our evaluation: Qwen3-8B [Yang et al., 2024] and DeepSeek-R1-Distil-Qwen-14B. We use YaRN scaling across our experiments to enable up to 128K context length [Peng et al., 2023]. We use the recommended decoding settings for both models, which is to use a temperature of 0.6 and a top-p value of 0.95, and to set the top-k value to 20 for Qwen3-8B. We report average results across 3 trials for all experiments.

We evaluate our approach using token budgets of 128 and 512. We fix the block size for clustering as $W = 8K$ tokens throughout our evaluation and set $\alpha = \frac{1}{2}W$, and we set $L = 128$ (meaning that we keep the last 128 to 256 tokens unclustered and we update the clusters every 128 decoding steps). We set the number of $k$-means iterations as 10 for prefill and 3 for refinement with our fast cluster update, and we use random initialization for the centroids. We also leave the first 10 "Attention Sink" tokens untouched due to their disproportionate importance [Xiao et al., 2023].

We compare our method with Squeezed Attention [Hooper et al., 2024a] and QUEST [Tang et al., 2024] as representative baselines for sparse attention. Note that for Squeezed Attention, we report results that leverage our optimized online clustering implementation in order to identify important KV cache entries among the generated output tokens (as well as windowed RoPE for improved key clustering), but without our method for approximating attention to clusters of tokens using the attention to the centroids. The token budgets for the baselines are adjusted to ensure that they retain the same number of tokens when factoring in the local buffer used by MULTIPOLE ATTENTION. For MULTIPOLE ATTENTION, we use one centroid per 16 tokens, and for QUEST we use a page size of 16. For Squeezed Attention, we use one centroid per 8 tokens for fair comparison, since it only

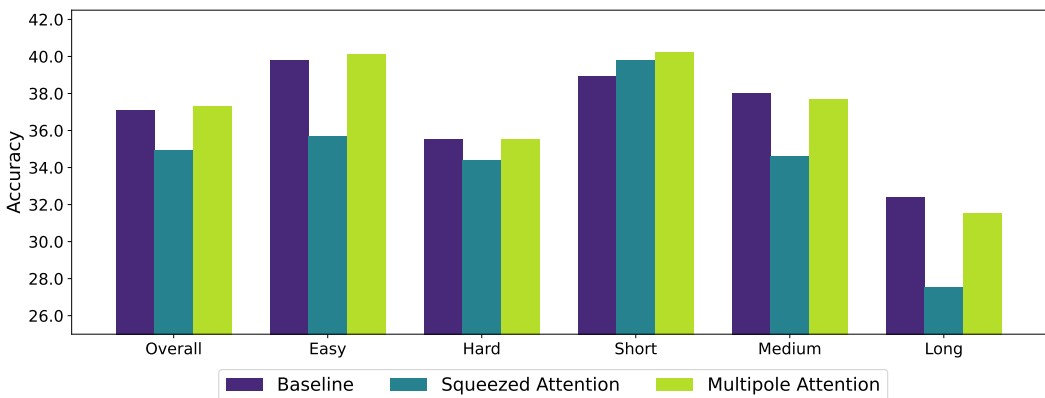

Figure 4: LongBenchV2 evaluation for DeepSeek-R1-Distil-Qwen-14B. We report accuracy on the full dataset, as well as easy/hard difficulty splits, and for short/medium/long (<32K/32K-128K/128K+) splits. We report accuracy for a token budget of 512, where we observe that MULTIPOLE ATTENTION (MpAttn) can achieve higher accuracy than Squeezed Attention (SqAttn) for the same token budget, and we observe no accuracy drop on the overall benchmark score.

Table 1: Accuracy on LongBench-V2 for Qwen3-8B with different token budgets. We report accuracy on the "Short" (<32K) split from the dataset. We include baseline comparisons with QUEST [Tang et al., 2024] as well as Squeezed Attention [Hooper et al., 2024a].

| Method | Token Budget = 128 | Token Budget = 512 |
|---|---|---|
| Baseline | 41.5 | 41.5 |
| QUEST | 13.7 | 22.2 |
| Squeezed Attention | 36.3 | 40.2 |
| **MULTIPOLE ATTENTION** | **38.1** | **40.4** |

has to store key centroids (whereas our method stores key and value centroids). For evaluation with the hierarchical variant of MULTIPOLE ATTENTION, we use a two-level lookup with one centroid per 64 tokens for the first level, and with one centroid per 8 tokens for the second level (and when comparing with the hierarchical centroid lookup from Squeezed Attention, we run their method with twice as many centroids at each level for fair comparison with our approach).

For our kernel benchmarking experiments, we use split size of 2048 in our FlashDecoding baseline and sparse FlashDecoding kernels, as well as the centroid lookup and replacement kernels. Note that this is suboptimal for a batch size of 1 for the lookup and replacement kernels, as the number of centroids is too small to saturate the GPU; however, we fixed this across all batch sizes for simplicity. We benchmark our kernel implementations using `triton.testing.do_bench` with 500 warmup runs and 500 measurement runs.

## 5.2 Evaluation

Figure 4 presents evaluation on the LongBenchV2 dataset for the Deepseek-R1-Distil-Qwen-14B model. These results show how MULTIPOLE ATTENTION is able to preserve the accuracy of the baseline, even with aggressive sparsity settings. Additionally, these results highlight how our method outperforms existing sparse attention methods, demonstrating how our multipole attention approximation is able to retain closer accuracy to the baseline by leveraging the scores to the centroids. Table 1 also presents evaluation on LongBenchV2 for the Qwen3-8B model. We include baseline comparisons with QUEST [Tang et al., 2024] and Squeezed Attention, demonstrating how MULTIPOLE ATTENTION outperforms existing sparse attention methods at different token budgets. Note that we only report accuracy on the "Short" (<32K) split from the dataset, since the Qwen3-8B model achieves lower than 25% accuracy on the medium / long splits (which is worse than random since the questions are multiple choice with 4 answers), primarily due to issues with endless repetitions. Appendix B provides additional analysis on LongBenchV1 [Bai et al., 2023] using a non-reasoning long context model, demonstrating how our method is generalizable for long context tasks outside of reasoning. Appendix C also provides ablations for the block size and number of cluster centroids hyperparameters.

Table 2: GSM-Infinite performance of Qwen3-8B at 8K and 16K context lengths. We measure the accuracy for 1-operation and 2-operation splits with the symbolic task. We report results for MULTIPOLE ATTENTION ("MpAttn") as well as Squeezed Attention [Hooper et al., 2024a] ("SqAttn"). We also include results using the hierarchical extension of our methodology, as well as the hierarchical lookup approach from Squeezed Attention (hierarchical configurations are marked with "-H"). We report the portion of memory operations required relative to the baseline (accounting for metadata overhead and assuming 8K/16K context lengths). MULTIPOLE ATTENTION achieves higher accuracy for the same memory overhead relative to Squeezed Attention across all token budgets and context lengths.

(a) Context length = 8K

| Method | Token Budget | Memory Operations | Acc. (1 op.) | Acc. (2 op.) |
|---|---|---|---|---|
| Baseline | - | 1 | 0.76 | 0.23 |
| SqAttn | 128 | 0.11 | 0.58 | 0.16 |
| MpAttn | 128 | 0.11 | **0.72** | **0.19** |
| SqAttn-H | 128 | 0.08 | 0.65 | 0.17 |
| MpAttn-H | 128 | 0.08 | **0.71** | **0.23** |
| SqAttn | 512 | 0.16 | 0.65 | 0.17 |
| MpAttn | 512 | 0.16 | **0.82** | **0.22** |

(b) Context length = 16K

| Method | Token Budget | Memory Operations | Acc. (1 op.) | Acc. (2 op.) |
|---|---|---|---|---|
| Baseline | - | 1 | 0.65 | 0.28 |
| SqAttn | 128 | 0.09 | 0.28 | 0.13 |
| MpAttn | 128 | 0.09 | **0.40** | **0.17** |
| SqAttn-H | 128 | 0.05 | 0.28 | 0.09 |
| MpAttn-H | 128 | 0.05 | **0.41** | **0.21** |
| SqAttn | 512 | 0.11 | 0.41 | 0.18 |
| MpAttn | 512 | 0.11 | **0.61** | **0.30** |

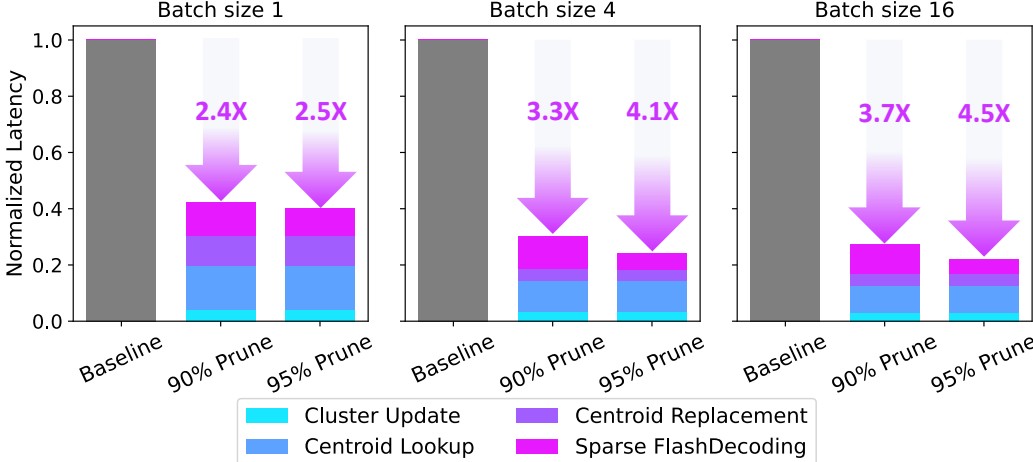

Figure 5: Attention runtime evaluation on an A6000 GPU for the Qwen3-8B model. We report results normalized to the FlashDecoding baseline. We show results for 90% and 95% sparsity (with one centroid per 16 tokens). We can attain up to 4.5× speedup relative to the baseline FlashDecoding kernels by leveraging MULTIPOLE ATTENTION.

Table 2 provides evaluation for the Qwen3-8B model on GSM-Infinite. We include results for the single-level and hierarchical variants of our method and the Squeezed Attention baseline [Hooper et al., 2024a]. We report results for 1 and 2 operation difficulty levels (referring to the number of operations required to get to the final answer). Our method retains closer to baseline accuracy than sparse attention methods like Squeezed Attention. The advantages of MULTIPOLE ATTENTION are particularly pronounced for aggressive sparsity regimes since there is a greater chance that sparse attention methods like Squeezed Attention will discard tokens which are important for retaining accuracy, whereas our approach always ensures that we retain at least an approximate representation for these tokens. Additionally, our hierarchical multipole algorithm attains higher accuracy for the same number of memory operations required relative to the one-level approach by reducing the overhead of the centroid lookup and centroid replacement operations.

## 5.3 System Benchmarking

We benchmarked our custom kernel implementations on both A6000 and A100 GPU platforms. Figure 5 shows the observed attention speedups on an A6000 GPU with our method, and kernel benchmarking evaluation on an A100 GPU is also provided in Appendix E. We observe up to

$2.5\times$, $4.1\times$, and $4.5\times$ speedups for batch sizes of 1, 4, and 16, respectively. We also show the breakdown of the portion of the runtime spent on our initial centroid comparison ("Centroid Lookup"), attention approximation using the centroids ("Centroid Replacement"), sparse Flash Decoding for exact attention with the important keys ("Sparse FlashDecoding"), as well as the median overhead from the fast cluster update ("Cluster Update"). For larger batch sizes, the centroid lookup and sparse FlashDecoding runtimes are substantial relative to the attention approximation kernel and cluster update, demonstrating the low overhead of our centroid replacement method relative to sparse attention, and how our fast cluster update enables quickly re-building an index over the generated tokens. Appendix D compares the attention speedups from MULTIPOLE ATTENTION with Squeezed Attention, demonstrating that we achieve comparable speedups for the same memory footprint. Appendix F provides additional analysis of the clustering overheads during prefill as well as decode.

## 6 Conclusion

Reasoning has emerged as a key enabler of complex problem solving capabilities in LLMs. While this reasoning through long chain-of-thought decoding is critical for achieving high accuracy on complex tasks, it comes with substantial efficiency costs due to the need to generate thousands of tokens before producing an answer. While sparse attention methods can accelerate decoding by reducing the memory bandwidth requirements for loading the KV cache, these approaches lead to substantial accuracy loss with aggressive sparsity regimes. Our work addresses these challenges by introducing MULTIPOLE ATTENTION, which reduces the KV cache memory requirements by only loading a small number of entries for exact attention computation, while maintaining approximate representations for the rest of the tokens. Our algorithm first clusters the keys based on semantic similarity, and then uses the corresponding cluster centroids both to select important key vectors and to approximate the attention to less important keys, thereby retaining important contextual information from the full sequence. We also design a fast cluster update strategy to facilitate quickly re-clustering the input to incorporate the newly generated tokens, which allows us to accelerate attention to the previous output tokens. We evaluate our method on emerging reasoning models and hard long-context reasoning tasks, demonstrating that our approach can maintain accuracy on complex reasoning tasks with aggressive KV cache sparsity settings. We also present efficient kernel implementations, demonstrating how our algorithm can accelerate attention by up to $4.5\times$ for long context reasoning applications.

## 7 Limitations

One limitation of our method is that it focuses on speeding up generation, and it does not accelerate attention during the prefill phase. One potential avenue for future work would be extending our method to support accelerating prefill. Another limitation is that additional implementation effort would be required to support our method in existing LLM serving frameworks. A third limitation is that our method requires additional memory to store the centroids (although this overhead is small relative to the size of the full KV cache).

## 8 Acknowledgements

We acknowledge gracious support from the FuriosaAI team including Jihoon Yoon, Suyeol Lee, and Hyung Il Koo, as well as from Intel, Apple, NVIDIA, and Mozilla. We also appreciate the support from Microsoft through their Accelerating Foundation Model Research, including great support from Sean Kuno. Furthermore, we appreciate support from Google Cloud, the Google TRC team, and specifically Jonathan Caton, and Prof. David Patterson. Prof. Keutzer's lab is sponsored by the Intel corporation, UC Berkeley oneAPI Center of Excellence, Intel VLAB team, as well as funding through BDD and BAIR. We appreciate great feedback and support from Ellick Chan, Saurabh Tangri, Andres Rodriguez, and Kittur Ganesh. Sehoon Kim would like to acknowledge the support from the Korea Foundation for Advanced Studies (KFAS). Michael W. Mahoney would also like to acknowledge a J. P. Morgan Chase Faculty Research Award as well as the DOE, NSF, and ONR. This work was supported by the Director, Office of Science, Office of Advanced Scientific Computing Research, of the U.S. Department of Energy under Contract No. DE-AC02-05CH11231. Our conclusions do not necessarily reflect the position or the policy of our sponsors, and no official endorsement should be inferred.

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

# A Extended Related Work: Only Loading Important KV Cache Entries

One previous approach for sparse attention has aimed to retain the full KV cache, but only load in the required KV cache entries for each decoding step. A common method that these approaches have used to identify important KV cache entries is clustering the keys. QUEST [Tang et al., 2024] grouped consecutive KV cache entries and derived representative vectors for the keys within each group. Their approach performs retrieval by comparing the current query with the representative vectors in order to identify whether the group is important and needs to be loaded for exact attention computation. While this method grouped consecutive keys, positional proximity in the sequence is not necessarily indicative of the similarity of the corresponding key vectors. Squeezed Attention [Hooper et al., 2024a] instead clustered keys based on semantic similarity, thereby allowing for precise identification of tokens which were likely to be high-scoring. Tactic [Zhu et al., 2025] also clustered the keys based on semantic similarity and then performed distribution fitting to help predict important KV cache entries.

Multiple prior works have framed the query-key comparison as a retrieval problem in order to leverage vector search methods to accelerate attention. One such method, PQCache [Zhang et al., 2024a], performed product quantization in order to perform vector search to select important keys, and then only computed exact attention using the selected keys. $A^2ATS$ [He et al., 2025] performed query-aware vector quantization in order to allow for accurate retrieval of important KV cache entries. RetrievalAttention [Liu et al., 2024a] constructs a vector index over the keys in order to facilitate retrieval of important keys, and offloads the full KV cache and the vector index to the CPU to perform the lookup.

In contrast with prior work on accelerating decoding through sparsely loading the KV cache, our method aims to select and load a small subset of important tokens for exact attention computation, while maintaining an approximate representation for the remaining KV cache entries. Our approach uses clustering to group semantically similar keys, followed by a fast centroid lookup that identifies important keys. Notably, our key centroids also serve as representative vectors for the keys in each of the clusters, and can therefore be used to approximate the attention to the keys which are not retrieved by the centroid lookup. This allows our method to maintain high accuracy even with aggressive sparsity settings (and for tasks which are sensitive to KV cache pruning).

Another related prior work is [Kang et al., 2023], which utilized a Fast Multipole Method (FMM)-inspired approach for accelerating the attention computation in Transformers. FMM [Coifman et al., 1993] is a related technique for accelerating simulations with N-Body problems which approximates groups of point masses as a single point mass and performs coarser grained approximations for masses which are further away from the current point of interest. This approach is analogous to our algorithm, which approximates keys using progressively coarser-grained key centroids as we get further from the current query token. However, unlike [Kang et al., 2023] which grouped KV cache entries based on positional proximity in the sequence, our method clusters keys based on semantic similarity, and therefore determines the granularity of approximation based on the *importance* of KV cache entries for the current query.

# B Additional Long Context Experiments

To demonstrate the broader applicability of our methodology on long context tasks other than reasoning, we evaluate our method on LongBenchV1 [Bai et al., 2023] using a non-reasoning long context model (Llama-3.1-8B-Instruct [Meta, 2024]), with results provided in Table 3. We also provide additional baseline comparisons with Squeezed Attention [Hooper et al., 2024a], DuoAttention [Xiao et al., 2024] and TOVA [Oren et al., 2024]. We use the evaluation setup from the DuoAttention open-source code, which simulates decoding for the last 50 prompt tokens, and we use a temperature of 0. We configure MULTIPOLE ATTENTION to use 95% sparsity and one centroid per 16 tokens, and then adjust the other methods to have equivalent KV cache budgets (88.75% sparsity when accounting for metadata). Our results demonstrate how our method substantially outperforms Squeezed Attention, DuoAttention, and TOVA on a range of long context evaluation tasks, and also how our method generalizes to long context tasks outside of reasoning.

Table 3: MULTIPOLE ATTENTION ("MpAttn") evaluation on non-synthetic tasks from LongBenchV1 [Bai et al., 2023] for the Llama-3.1-8B-Instruct model. We include baseline comparisons with Squeezed Attention ("SqAttn") [Hooper et al., 2024a] as well as DuoAttention ("DuoAttn") [Xiao et al., 2024] and TOVA [Oren et al., 2024] (configured to have the same KV cache budget as our method). We also provide estimates for the number of memory operations required for KV cache loading, including metadata (normalized to the number of KV cache memory operations required for the baseline). Note that the TOVA algorithm allows each query head to select different KV entries and would therefore have a higher KV footprint for GQA models even for the same sparsity settings. MULTIPOLE ATTENTION provides noticeable accuracy improvements for the same KV cache budget relative to existing methods.

| Config | Mem. Ops. | Single-Doc. QA | | | Multi-Doc. QA | | | Summarization | | | Few-shot Learning | | | Code | | Avg. |
| | | NQA | Qspr | MFQA | HPQA | 2Wiki | MSQ | GRep | QMSM | MNews | TREC | TQA | SSum | RB | LCC | |
|---|---|---|---|---|---|---|---|---|---|---|---|---|---|---|---|---|
| Baseline | 1.00 | 30.1 | 45.3 | 55.5 | 56.0 | 44.8 | 30.5 | 35.2 | 25.3 | 27.3 | 72.5 | 91.2 | 43.2 | 49.0 | 52.5 | 47.0 |
| DuoAttn | 0.1125 | 13.2 | 16.8 | 21.3 | 29.1 | 17.4 | 8.2 | 20.5 | 17.3 | 20.1 | 32.0 | 52.4 | 36.8 | 47.0 | **49.0** | 27.2 |
| TOVA | 0.1125 | 24.1 | 14.0 | 23.2 | 41.5 | 21.1 | 15.1 | 25.5 | 19.9 | 20.9 | 35.0 | 86.1 | 39.9 | 49.1 | 48.6 | 33.1 |
| SqAttn | 0.1125 | 29.3 | 24.2 | 38.6 | 50.1 | **35.2** | 27.0 | 31.5 | 24.5 | 18.9 | 55.0 | 81.6 | 41.5 | **49.9** | 47.4 | 39.6 |
| **MpAttn** | 0.1125 | **31.6** | **32.9** | **42.0** | **54.3** | 31.3 | **30.3** | **32.8** | **24.7** | **22.9** | **67.5** | **86.7** | **42.1** | 46.8 | **48.7** | **42.5** |

## C    Hyperparameter Ablations

Tables 4 and 5 provide ablations for the number of centroids as well as block size, respectively. We report LongBenchV2 accuracy for the Qwen3-8B model on the "short" (<32K context length) split with a token budget of 128. We find that if the number of KV tokens per centroid is increased or if the block size is decreased, the accuracy is degraded. However, increasing these parameters leads to additional inference costs. We therefore used 1 centroid per 16 tokens and a block size of 8K to retain model accuracy without introducing substantial memory and latency overheads.

Table 4: LongBenchV2 accuracy on the "Short" (<32K) split versus ratio of centroids to KV tokens for the Qwen3-8B model. We report results for a token budget of 128. The configuration used throughout the paper is bolded.

| Baseline | Ratio=1/128 | Ratio=1/64 | Ratio=1/32 | **Ratio=1/16** | Ratio=1/8 |
|---|---|---|---|---|---|
| 41.5 | 29.3 | 35.4 | 38.5 | **38.1** | 37.0 |

Table 5: LongBenchV2 accuracy on the "Short" (<32K) split versus clustering block size $W$ for the Qwen3-8B model. We report results for a token budget of 128. The configuration used throughout the paper is bolded.

| Baseline | $W = 2K$ | $W = 4K$ | **$W = 8K$** | $W = 16K$ |
|---|---|---|---|---|
| 41.5 | 35.0 | 38.0 | **38.1** | 36.9 |

## D    Runtime Comparison with Squeezed Attention

In Table 6, we compare the latency for MULTIPOLE ATTENTION versus Squeezed Attention [Hooper et al., 2024a] for the same memory footprint (using one centroid per 16 tokens for MULTIPOLE ATTENTION and one centroid per 8 tokens for Squeezed Attention, as outlined in Section 5.1). Note that the Squeezed Attention baseline we compare with in our experiments also contains the clustering improvements from MULTIPOLE ATTENTION, which are required to allow it to be applied for tasks where the prefill is only known at runtime and during the generation process to accelerate attention to newly generated tokens (as the baseline Squeezed Attention method cannot be run online). We report the speedups for Squeezed Attention and MULTIPOLE ATTENTION on an A6000 GPU with batch sizes of 1/4/16 (assuming 90% sparsity). These results outline how MULTIPOLE ATTENTION provides similar latency for the same sparsity level as Squeezed Attention, while providing substantial accuracy improvements (as highlighted in Section 5.2).

Table 6: Attention speedup for MULTIPOLE ATTENTION and Squeezed Attention [Hooper et al., 2024a] for the Qwen3-8B model on an A6000 GPU, relative to the full attention baseline.

| Method | Batch Size = 1 | Batch Size = 4 | Batch Size = 16 |
|---|---|---|---|
| Squeezed Attention | 2.8× | 3.3× | 3.6× |
| MULTIPOLE ATTENTION | 2.4× | 3.3× | 3.7× |

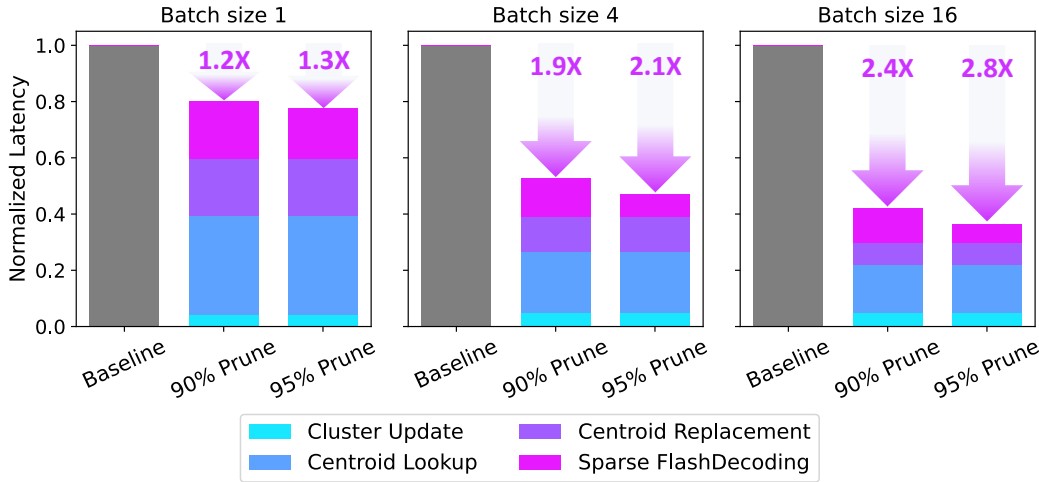

Figure 6: Attention runtime evaluation on an A100 GPU for the Qwen3-8B model. We report results normalized to the FlashDecoding baseline. We show results for 90% and 95% pruning, and for batch sizes of 1, 4, and 16 (and assuming one centroid per 16 tokens). We can attain up to 2.8× speedup relative to the baseline FlashDecoding kernels by leveraging MULTIPOLE ATTENTION.

## E  A100 Kernel Benchmarking

Figure 6 shows the observed attention speedups on an A100 GPU with our method. We observe up to 1.3×, 2.1×, and 2.8× speedups for batch sizes of 1, 4, and 16, respectively. Note that there are reduced speedups for batch size of 1 due to there being insufficient work to saturate the GPU.

## F  Clustering Runtime

Tables 7 and 8 provide measured runtime for the fast cluster update during generation on A6000 and A100 GPUs, respectively. We find that the overhead of the fast cluster update is typically 3-4% of the runtime of the FlashDecoding baseline on an A6000 GPU, and 5% of the runtime of the FlashDecoding baseline on an A100 GPU. Table 9 also reports the clustering overhead during prefill, which is relatively low (between 13-15% on A6000/A100 GPUs).

Table 7: Runtime (in milliseconds) for running our fast clustering update for the Qwen3-8B model on an A6000 GPU. The clustering runtimes for 8K and 12K tokens serve as the median and maximum clustering runtimes, respectively, since we set $W = 8K$ and $\alpha = \frac{1}{2}W$ throughout our evaluation. Since we update the clusters once every 128 decoding steps (and need to run FlashDecoding at each step), clustering adds a median of 3-4% overhead and a maximum of 6% overhead during the decoding process.

| Batch Size | FlashDecoding Runtime (128K context length) | Clustering Runtime (8K) | Clustering Runtime (12K) |
|---|---|---|---|
| 1 | 0.8 | 4.1 | 6.6 |
| 4 | 2.9 | 12.4 | 23.1 |
| 16 | 11.6 | 47.6 | 89.8 |

Table 8: Runtime (in milliseconds) for running our fast clustering update for the Qwen3-8B model on an A100 GPU. The clustering runtimes for 8K and 12K tokens serve as the median and maximum clustering runtimes, respectively, since we set $W = 8K$ and $\alpha = \frac{1}{2}W$ throughout our evaluation. Since we update the clusters once every 128 decoding steps (and need to run FlashDecoding at each step), clustering adds a median of 5% overhead and a maximum of 8-9% overhead during the decoding process.

| Batch Size | FlashDecoding Runtime (128K context length) | Clustering Runtime (8K) | Clustering Runtime (12K) |
|---|---|---|---|
| 1 | 0.4 | 2.6 | 4.2 |
| 4 | 1.3 | 8.4 | 14.8 |
| 16 | 5.1 | 31.7 | 59.7 |

Table 9: Runtime overhead (in milliseconds) for running our blockwise clustering algorithm during prefill for the Qwen3-8B model. We report overheads on both A100 and A6000 GPUs for a batch size of 1, assuming 128K context length prefill (divided into 8K blocks). Clustering adds a relatively low overhead of between 13%-15% when processing the input prompt.

| GPU | FlashAttention Runtime (128K context length) | Clustering Runtime (16 blocks of size 8K) | Clustering Overhead (%). |
|---|---|---|---|
| A6000 | 1506 | 192 | 13 |
| A100 | 787 | 118 | 15 |

