# OpenReview forum: "Multipole Attention for Efficient Long Context Reasoning"
_NeurIPS.cc/2025/Conference — NeurIPS 2025 poster_

### Official Review · Reviewer_TWnq · 2025-06-23

**Clarity:** 2
**Significance:** 3
**Originality:** 3
**Rating:** 4
**Confidence:** 3

**Summary:**

This paper proposes Multipole Attention, a method to speed up long-chain reasoning in Large Reasoning Models (LRMs) by computing exact attention only for key tokens and approximating the rest. It clusters semantically similar key vectors and uses their centroids to guide attention, balancing efficiency and accuracy. A fast cluster update mechanism enables efficient handling of long sequences during generation. Evaluations on models like Qwen-8B and Deepseek-R1 show up to 4.5× speedup with minimal accuracy loss on complex reasoning tasks.

**Questions:**

N/A

**Ethical Concerns:**

["NO or VERY MINOR ethics concerns only"]

**Final Justification:**

Recommended score: 4

 Justification: Considering the novelty of the paper and the sufficiency of the experiments, I believe the minor writing issues do not undermine the strength of the work. Therefore, I recommend a score of 4.

**Limitations:**

yes

**Quality:**

3

**Strengths And Weaknesses:**

Strengths:
1. The motivation of this paper is apparent and addresses a practical need, especially given the growing use of LRMs in scenarios where they need to generate very long outputs. The proposed method has high application value in such contexts.
2. This paper introduces the idea of applying clustering techniques to the attention mechanism, which is a relatively novel and innovative approach.
3. The experiments in this paper are comprehensive and effectively support the paper’s main claims.

Weaknesses:
1.  The paper is not well-written; it contains a large amount of textual description but lacks formal mathematical formulations, which makes it difficult to follow and understand.
2. In line 179 of the "Efficient Online Clustering" section, the term "BS Tokens" is used without any prior explanation or background, which makes the paper less reader-friendly and harder to follow.
3. As shown in Figure 4, the proposed method shows the largest performance gap compared to the baseline on medium-length sequences. The authors attribute this to the issue of endless repetition. However, they do not provide further analysis or validate this explanation on datasets that do not suffer from repetition issues.

---

> ### Author Rebuttal · Authors · 2025-07-31
>
> > R4-1: The paper is not well-written; it contains a large amount of textual description but lacks formal mathematical formulations, which makes it difficult to follow and understand.
>
> We appreciate the feedback from the reviewer. For the final version of the paper, we will ensure that we include relevant mathematical formulations in addition to textual descriptions in order to ensure the methodology is described clearly.
>
>
> > R4-2: In line 179 of the "Efficient Online Clustering" section, the term "BS Tokens" is used without any prior explanation or background, which makes the paper less reader-friendly and harder to follow.
>
> We appreciate the reviewer’s feedback with respect to the writing. Line 179 (`We perform clustering in blocks of BS tokens`) was intended to define BS as a variable corresponding to the number of tokens in each block. We will rework this sentence to enhance clarity.
>
>
> > R4-3: As shown in Figure 4, the proposed method shows the largest performance gap compared to the baseline on medium-length sequences. The authors attribute this to the issue of endless repetition. However, they do not provide further analysis or validate this explanation on datasets that do not suffer from repetition issues.
>
> We would like to kindly note that we did not attribute the performance gap to endless repetitions. The repetition issue was observed with the Qwen3-8B model when running on medium / long context length inputs for the LongBenchV2 dataset. This is a known issue with the model when not using recommended decoding settings (as mentioned in the model page [1]); however, for long context inputs, we observed this even when using recommended decoding settings. As such, we did not include Qwen3-8B evaluation for medium/long context length splits in the paper.
>
> [1] https://huggingface.co/Qwen/Qwen3-8B

---

> > ### Comment · Reviewer_TWnq · 2025-08-05
> > **Response to author**
> >
> > Thank you for the authors' response. Since my score is already quite high, I will keep it as it is.

---

### Official Review · Reviewer_vJGV · 2025-07-02

**Clarity:** 3
**Significance:** 3
**Originality:** 3
**Rating:** 4
**Confidence:** 3

**Summary:**

This paper introduces a method that uses hierarchical key clustering and centroid-based attention to reduce the KV cache while preserving contextual information. To accelerate attention computation, the authors propose a Fast Online Clustering algorithm and implement system-level optimizations using a Triton kernel.

**Questions:**

How does the method perform across different context lengths? An ablation study or detailed analysis would be helpful.

**Ethical Concerns:**

["NO or VERY MINOR ethics concerns only"]

**Final Justification:**

This paper is well-motivated and provides solid experiments. All my concerns have been addressed; therefore, I will maintain my positive score.

**Limitations:**

See Weakness

**Quality:**

3

**Strengths And Weaknesses:**

Strengths:
1. The idea of using clustering to compress the KV cache is well-motivated, and the hierarchical clustering design aligns well with modern memory access patterns;
2. The proposed Efficient Online Clustering algorithm makes KV cache compression practically feasible;
3. The system implementation based on Triton kernels demonstrates the efficiency of Efficient Online Clustering convincingly.

Weaknesses:
1. The experiments are limited to only two benchmarks: LongBenchV2 and GSM-Infinite. The evaluation lacks diversity in tasks and datasets.
2. The comparison is also restricted, focusing solely on Squeezed Attention and QUEST.
3. It would strengthen the paper to include additional benchmarks, such as Needle-in-a-Haystack [1], and comparisons with more recent methods like DuoAttention [2] and TOVA [3].

References:

[1] Needle-in-a-Haystack: https://github.com/gkamradt/LLMTest_NeedleInAHaystack

[2] DuoAttention: Efficient Long-Context LLM Inference with Retrieval and Streaming Heads

[3] Transformers are Multi-State RNNs

---

> ### Author Rebuttal · Authors · 2025-07-31
>
> > R3-1: The experiments are limited to only two benchmarks: LongBenchV2 and GSM-Infinite. The evaluation lacks diversity in tasks and datasets. The comparison is also restricted, focusing solely on Squeezed Attention and QUEST. It would strengthen the paper to include additional benchmarks, such as Needle-in-a-Haystack [1], and comparisons with more recent methods like DuoAttention [2] and TOVA [3].
>
> We appreciate the reviewer’s feedback. In our submission, we evaluated on a range of tasks requiring complex reasoning capabilities; LongBenchV2 includes questions from a range of downstream tasks, including single and multi-document QA, in-context learning, and code repository understanding, and GSM-Infinite contains arithmetic reasoning questions.
>
> To bolster the evaluation from our paper, we have conducted additional evaluation on the LongBenchV1 benchmark suite and passkey retrieval task to demonstrate the generalizability of our method. We have included additional comparisons with DuoAttention and TOVA below, which we will include in the final version of our paper. We compare accuracy on LongBenchV1 for the Llama-3.1-8B-Instruct model using the evaluation setup from the open-source DuoAttention code which simulates decoding for the final 50 prompt tokens. We configure Multipole Attention to use 95% sparsity and 1 centroid per 16 tokens, and then adjust the other methods to have equivalent KV cache budgets (around 11.25% of the full KV cache). Our results demonstrate how our method substantially outperforms DuoAttention and TOVA, and also how our method generalizes to non-reasoning models and other long context tasks. Note that the TOVA algorithm allows each query head to select different keys and values and therefore has a larger KV footprint for GQA models even for the same sparsity settings.
>
> | Method                     | Baseline Accuracy | DuoAttention | TOVA | Squeezed Attention | **Multipole Attention** |
> | -------------------------- | ----------------- | ------------ | ---- | ------------------ | ------------------- |
> | Average LongBench Accuracy | 47.8              | 26.2         | 34.3 | 41.0               | **43.7**                |
>
>
> *Average LongBenchV1 accuracy (Llama-3.1-8B-Instruct, evaluated on English tasks) for DuoAttention, TOVA, Squeezed Attention, and Multipole Attention.*
>
> Additionally, we have provided needle-in-a-haystack evaluation for our method using the evaluation code from DuoAttention (sweeping across different context lengths and depths). We ran evaluation for the baseline as well as Squeezed Attention and Multipole Attention with the Llama-3.1-8B-Instruct model using 128 token budget. We found that the accuracy for Multipole Attention / Squeezed Attention were equal / nearly equal to the baseline accuracy, even with aggressive KV cache compression. Note that while Multipole Attention retains the accuracy of the baseline model, the benefits relative to Squeezed Attention (and other prior sparse attention methods) are more pronounced for tasks which require more broad contextual information from across the input context and previously generated tokens (e.g. complex reasoning tasks). We will include the full visualizations across different context lengths and depths in the final version of our paper.
>
> | Method              | Overall Accuracy |
> | ------------------- | ---------------- |
> | Baseline            | 0.912            |
> | Squeezed Attention  | 0.911            |
> | **Multipole Attention** | **0.912**            |
>
> *Passkey retrieval accuracy for Llama-3.1-8B-Instruct model (token budget 128).*
>
>
> > R3-3: How does the method perform across different context lengths? An ablation study or detailed analysis would be helpful.
>
> We show results in Figure 4 which outlines how Multipole Attention performs across different context length splits for document processing tasks in LongBenchV2 (short/medium/long, i.e. <32K/32K-128K/128K+ context lengths)). Looking at the configuration with token budget of 512, we observe comparable performance to the baseline across all 3 context length splits. Additionally, Table 2 highlights the performance of Multipole Attention for both 8K and 16K context length inputs with arithmetic computation tasks (GSM-Infinite). We find that Multipole Attention provides consistent accuracy advantages relative to Squeezed Attention across different context lengths across both document processing tasks (LongBenchV2) as well as arithmetic reasoning tasks (GSM-Infinite), and retains close to the baseline accuracy even with aggressive KV cache compression budget.

---

> > ### Comment · Reviewer_vJGV · 2025-08-05
> > **keep my positive score**
> >
> > Thank you to the author for the experiments. The experimental results show that Multipole Attention works very well on multiple tasks, and I will keep my positive score.

---

### Official Review · Reviewer_zdAu · 2025-07-03

**Clarity:** 2
**Significance:** 2
**Originality:** 2
**Rating:** 4
**Confidence:** 3

**Summary:**

This paper proposes a method to reduce KV cache requirements during long-context inference by performing exact attention only for tokens identified as important, while using approximate cluster representations for the remaining tokens. Attention for these other tokens is computed with respect to the cluster centroids. This approach reduces KV cache demands while maintaining a reasonably good representation over the sequence and preserving final accuracy fairly well.

**Questions:**

How does Multipole Attention compare to Squeezed Attention when both accuracy and runtime are jointly taken into account? In particular, is it possible to report results in a way that clarifies the trade-off between these two dimensions?

**Ethical Concerns:**

["NO or VERY MINOR ethics concerns only"]

**Final Justification:**

Clarifications from the authors about the evaluation setup convinced me to update the overall assessment.

**Limitations:**

yes

**Quality:**

3

**Strengths And Weaknesses:**

## Strengths

The paper introduces a potentially useful method to speed up long-context inference while preserving accuracy to a reasonable degree. The algorithmic design and its description appear sound. The system-level implementation make likely to be practical in real-world scenarios.

## Weaknesses

The contribution is conceptually an incremental improvement over Squeezed Attention. In practice, while it appears to preserve accuracy slightly better than squeezed attention, it also seems more computationally expensive, since it requires computing approximate attention over cluster centroids.

In general, the comparison with Squeezed Attention could be performed along two dimensions simultaneously: accuracy preservation and runtime. This would allow us to see whether the proposed approach is significantly better for a given runtime budget, or significantly faster for a given target accuracy.

Regarding Figure 4: “We report accuracy for a token budget of 512, where we observe that MULTIPOLE ATTENTION (MpAttn) can achieve higher accuracy than Squeezed Attention (SqAttn) for the same token budget.” This choice of token budget appears somewhat cherry-picked.

Overall, it would be more convincing if the experimental evaluation was more thorough and systematic.

Finally, there is no measure of uncertainty reported in the tables and figures (e.g., standard deviation or confidence intervals).

Minor: The citations are not correctly formatted (\citet, \citep, etc.)

---

> ### Author Rebuttal · Authors · 2025-07-31
>
> > R2-1: The contribution is conceptually an incremental improvement over Squeezed Attention. In practice, while it appears to preserve accuracy slightly better than squeezed attention, it also seems more computationally expensive, since it requires computing approximate attention over cluster centroids. In general, the comparison with Squeezed Attention could be performed along two dimensions simultaneously: accuracy preservation and runtime. This would allow us to see whether the proposed approach is significantly better for a given runtime budget, or significantly faster for a given target accuracy.
> > How does Multipole Attention compare to Squeezed Attention when both accuracy and runtime are jointly taken into account? In particular, is it possible to report results in a way that clarifies the trade-off between these two dimensions?
>
> We appreciate the feedback. **As highlighted below, we can achieve 43-54% less accuracy degradation from the baseline than SqueezeAttention when comparing configurations with the same runtime.** To attain these benefits, we made multiple improvements relative to Squeezed Attention:
>
> - We incorporate algorithmic improvements by approximating attention to less important KV cache tokens, and we develop a systems implementation to compute this approximate attention to less important regions with minimal overhead
> - We design an efficient clustering strategy such that we can apply clustering online during inference (whereas Squeezed Attention could only be applied for tasks where the prefill was available offline ahead of time), and we also design a fast cluster update procedure such that we can also apply clustering to the newly generated tokens during long generation
>
> In the paper, we provided comparisons which used memory footprint as a proxy for runtime, and demonstrated how Multipole Attention provides improved accuracy relative to Squeezed Attention for the same memory footprint. **For example, Table 2 highlights how the best Multipole Attention configuration with token budget 128 achieves 56-96% less accuracy degradation from the baseline than the best Squeezed Attention configurations with similar memory footprint (across both context lengths and 1 and 2-operation tasks).** Here, we also provide additional analysis to clarify how the reduced memory footprint correlates with a reduction in runtime, which we will incorporate into the final version of our paper.
>
> We compare the accuracy with Multipole Attention and Squeezed Attention for the Qwen3-8B model on GSM-Infinite (16K context length) and LongBenchV2 with a 128-token budget and using one centroid per 16 tokens (and using 2x more centroids for Squeezed Attention to ensure the same memory footprint, since Multipole Attention requires computing centroids for the keys and values). Note that the Squeezed Attention baseline we compared with in our experiments also contains the clustering improvements from Multipole Attention, which are required to allow it to be applied for tasks where the prefill is only known at runtime and during the generation process to accelerate attention to newly generated tokens (as the clustering from the original Squeezed Attention method cannot be run online). We also report the speedups for Squeezed Attention and Multipole Attention on an A6000 GPU with batch sizes of 1/4/16 (assuming 90% sparsity). These results outline how Multipole Attention provides substantial accuracy improvements for a similar runtime speedup relative to Squeezed Attention (with 43-54% less accuracy degradation from the baseline across GSM-Infinite and LongBenchV2 datasets).
>
> |                     | GSM-Inf - 1-Op Accuracy | GSM-Inf - 2-Op Accuracy | LongBenchV2 Accuracy | Speedup (Batch size 1) | Speedup (Batch size 4) | Speedup (Batch size 16) |
> | ------------------- | ----------------------- | ----------------------- | -------------------- | ---------------------- | ---------------------- | ----------------------- |
> | Baseline            | 0.62                    | 0.28                    | 44.4                 | 1                      | 1                      | 1                       |
> | Squeezed Attention  | 0.27                    | 0.11                    | 36.1                 | 2.8                    | 3.3                    | 3.6                     |
> | **Multipole Attention** | **0.42**                    | **0.19**                    | **40.6**                 | **2.4**                    | **3.3**                    | **3.7**                     |
>
> *GSM-Infinite 1-op/2-op accuracy and LongBenchV2 accuracy versus attention speedup for Squeezed Attention and Multipole Attention with the Qwen3-8B model.*
>
>
>
> > R2-2: Regarding Figure 4: “We report accuracy for a token budget of 512, where we observe that MULTIPOLE ATTENTION (MpAttn) can achieve higher accuracy than Squeezed Attention (SqAttn) for the same token budget.” This choice of token budget appears somewhat cherry-picked.
>
> We would like to kindly note that we reported accuracy results for token budgets of 128 and 512 across both datasets (LongBenchV2 as well as GSM-Infinite) and for both models, and that Figure 4 contains results for a token budget of 128 and 512. Please see Table 1 and Figure 4 for LongBenchV2 results, and Table 2 for GSM-Infinite results.
>
>
> > R2-3: Finally, there is no measure of uncertainty reported in the tables and figures (e.g., standard deviation or confidence intervals).
>
> We appreciate the reviewer’s feedback. In the final version, we will incorporate experimental results with multiple samples with error bars.
>
>
> > R2-4: Minor: The citations are not correctly formatted (\citet, \citep, etc.)
>
> Thank you for the formatting correction - we will fix this in the final version of our paper.

---

> > ### Comment · Reviewer_zdAu · 2025-08-04
> >
> > Thank you for your answer and clarifications. I have updated and increased my assessment of the work.

---

### Official Review · Reviewer_zAWi · 2025-07-03

**Clarity:** 4
**Significance:** 4
**Originality:** 4
**Rating:** 5
**Confidence:** 2

**Summary:**

The paper introduces Multipole Attention, a sparse attention mechanism which selectively computes exact attention only for the most relevant tokens, while using approximate attention based on semantically clustered key-value centroids for the rest of the sequence. To achieve this, the method first performs semantic k-means clustering over key vectors and derives representative centroids. At inference time, the model compares the query with these centroids to determine which tokens to retrieve for exact attention and which to approximate. To ensure scalability during generation, the authors propose a fast online clustering algorithm with blockwise updates and a sliding window to re-cluster newly generated tokens efficiently. Additionally, the paper extends this to a hierarchical multipole formulation, enabling progressively coarser approximations for less important tokens. The authors implement this method using custom Triton kernels and demonstrate that it achieves up to 4.5× decoding speedup on reasoning benchmarks like LongBenchV2 and GSM-Infinite, while maintaining accuracy significantly better than prior sparse attention baselines such as QUEST and Squeezed Attention.

**Questions:**

Can multipole attention be applied to tasks that are not reasoning related?

**Ethical Concerns:**

["NO or VERY MINOR ethics concerns only"]

**Final Justification:**

Overall, I thought this was an interesting work that provides rigorous experimentation to demonstrate that the proposed multipole attention helps improve long context reasoning performance.  I appreciate that the authors performed additional experiments on other long context tasks outside of reasoning and additional hyperparameter ablations.  Because of this I recommend accept.

**Limitations:**

Yes

**Quality:**

3

**Strengths And Weaknesses:**

Strengths:
- introduces a novel sparse attention mechanism which can be used to substitute attention during inference
- experimental results on long reasoning benchmarks demonstrates that proposed method can retain strong reasoning performance, even with aggressive KV cache sparsity.
- proposed technique is designed to be efficient and scalable (authors propose using fast online clustering and have a custom triton kernel implementation)
- writing is clear

Weaknesses:
- generalization outside of reasoning- while the paper is focused on the reasoning task, I don't think the idea of being able to perform well on long contexts is restricted to reasoning.  It would be interesting to see how multipole attention performs in broader use cases such as summarizing/qa for long documents
- missing ablations for hyperparameters such as number of centroids and block size

---

> ### Author Rebuttal · Authors · 2025-07-31
>
> > R1-1: Generalization outside of reasoning- while the paper is focused on the reasoning task, I don't think the idea of being able to perform well on long contexts is restricted to reasoning. It would be interesting to see how multipole attention performs in broader use cases such as summarizing/qa for long documents
>
> We completely agree and appreciate the reviewer’s feedback. We focused our evaluation on reasoning models since newer reasoning models typically have much longer output generation lengths, which makes accelerating the generation phase particularly critical, and since reasoning tasks require broad understanding of the input context and experience greater accuracy loss with existing sparse attention methods [1]. However, note that we evaluated these reasoning models on a range of tasks requiring complex reasoning capabilities; LongBenchV2 includes questions from a range of downstream tasks, including single and multi-document QA, in-context learning, and code repository understanding, and GSM-Infinite contains arithmetic reasoning questions.
>
> Additionally, we would like to highlight that we have evaluated Multipole Attention on non-reasoning models (see response R3-1) on various long context length tasks, including the LongBench dataset (which contains single and multi-document QA, summarization, in-context learning, and code repository understanding) and passkey retrieval. These results demonstrate the generalizability of our method.
>
> [1] Can LLMs maintain fundamental abilities under KV cache compression?
>
>
> > R1-2: Missing ablations for hyperparameters such as number of centroids and block size
>
> We have included ablations here for the number of centroids as well as block size.  We report LongBenchV2 accuracy for the Qwen3-8B model on the “short” (<32K context length) split with a token budget of 128. For the number of centroids ablation, these results demonstrate that if we use coarser grained clustering than one centroid per 16 tokens (the main configuration used in the paper), the accuracy is noticeably degraded - we therefore used 1 centroid per 16 tokens as the main configuration throughout our work. For the block size ablation, we find that if we use less than 8K block size, the accuracy is degraded. If we use greater than 8K block size,  the efficiency overhead of clustering will be noticeably increased; we therefore selected 8K as the setting throughout our experiments.
>
> | Centroid / Token Ratio | Baseline | 1/128 | 1/64 | 1/32 | 1/16 | 1/8  |
> | ---------------------- | -------- | ----- | ---- | ---- | ---- | ---- |
> | LongBenchV2 Accuracy   | 44.4     | 31.1  | 35   | 36.1 | **40.6** | 38.3 |
>
> *LongBenchV2 accuracy versus ratio of centroids to KV tokens for Multipole Attention with the Qwen3-8B model (bold - the config used in the paper).*
>
> | Block Size           | Baseline | 2K   | 4K   | 8K   | 16K  |
> | -------------------- | ------- | ---- | ---- | ---- | ---- |
> | LongBenchV2 Accuracy | 44.4     | 35.0 | 38.3 | **40.6** | 36.7 |
>
> *LongBenchV2 accuracy versus clustering block size for Multipole Attention with the Qwen3-8B model (bold - the config used in the paper).*

---

> > ### Comment · Reviewer_zAWi · 2025-08-04
> >
> > Thank you for the rebuttal, I appreciate the additional experiments for other long context datasets and ablations for number of centroids.  I still have a positive view of this paper and will maintain my score of 5.

---

### Decision · Program_Chairs · 2025-09-17

**Decision:**

Accept (poster)

**Comment:**

This paper proposes a method to reduce KV cache requirements during long-context inference by performing exact attention only for tokens identified as important, while using approximate cluster representations for the remaining tokens.

Strengths:

1. The paper introduces a novel sparse attention mechanism, which has high application value in LRMs for long-contexts.
2. The algorithmic design and its description appear sound, being efficient and scalable (i.e., fast online clustering and a custom triton kernel implementation).
3. The experiments in this paper are comprehensive and effectively support the paper’s main claims.

Weakness:

1. generalization outside of reasoning tasks and some hyperparameters analysis missing (` zAWi `)
2. lack of detailed comparison with baseline Squeezed Attention (`zdAu`)
3. more benchmarks and baselines (`vJGV`)

All weakness have been addressed during rebuttal, and all reviewers consider this is a good paper. Please do revise it accordingly.